



# A benchmark dataset for global evapotranspiration estimation based on FLUXNET2015 from 2000 to 2022

Wangyipu Li[1,2], Zhaoyuan Yao[1,2], Yifan Qu[1,2], Hanbo Yang[3], Yang Song[4], Lisheng Song[5], Lifeng Wu[6], Yaokui Cui[1,2]

[1]Institute of RS and GIS, School of Earth and Space Sciences, Peking University, Beijing 100871, China
[2]Beijing Key Laboratory of Spatial Information Integration and Its Applications, Beijing 100871, China
[3]State Key Laboratory of Hydro-science and Engineering, Department of Hydraulic Engineering, Tsinghua University, Beijing 100084, China
[4]Institute of Crop Sciences, Chinese Academy of Agricultural Sciences, Beijing 100081, China
[5]Key Laboratory of Earth Surface Processes and Regional Response in the Yangtze–Huaihe River Basin, School of Geography and Tourism, Anhui Normal University, Wuhu 241000, China
[6]Faculty of Modern Agricultural Engineering, Kunming University of Science and Technology, Kunming 650500, China

*Correspondence to*: Yaokui Cui (yaokuicui@pku.edu.cn)

**Abstract.** Evapotranspiration (ET) is a crucial component of the terrestrial hydrological cycle. Latent heat flux (LE, equivalent to ET in $W/m^2$) observed by the eddy covariance (EC) technique, as known as $LE_{EC}$, has been publicly recognized as highly accurate benchmark for global ET estimation. Currently, there is an increasing need for long time-series benchmark data to support climate change analysis, construction of new models, and validation of new products. However, existing $LE_{EC}$ datasets, like FLUXNET2015, face significant challenges due to limited observation periods and extensive data gaps. This hinders their application. To address these issues, we developed a gap-filling and prolongation framework for $LE_{EC}$ data and established a benchmark dataset for global ET estimation from 2000 to 2022 across 64 sites at various time scales. The framework mainly contained 3 parts: site selection and data pre-processing, gap-filled half-hourly / hourly LE data generation, and prolonged daily LE data generation. We selected 64 sites from FLUXNET2015 based on a rigorous filtering criterion. A novel bias-corrected random forest (RF) algorithm was used as the gap-filling and prolongation algorithm of the framework to produce seamless half-hourly and daily LE data. After analysis, the framework using novel bias-corrected RF algorithm achieves excellent performance both in hourly gap-filling and daily prolongation, with a median RMSE of 32.84 $W/m^2$ and 16.58 $W/m^2$, respectively. The algorithm significantly improved the gap-filling performance for long gaps and extreme values compared with the original RF and marginal distribution sampling (MDS) algorithm. The results demonstrate robust prolongation performance of our framework both on prolonging directions and temporal stability. There is a high consistency in data distribution between our gap-filled dataset and FLUXNET2015 dataset. In conclusion, a benchmark dataset for global ET estimation based on FLUXNET2015 from 2000 to 2022 was firstly published. This dataset can strongly provide data support for ET modelling, water-carbon cycle monitoring and climate change analysis. It is made freely available via the following repository: https://doi.org/10.5281/zenodo.13853409 (Li et al., 2024b).



## 1 Introduction

Terrestrial evapotranspiration (ET), which represents the move and phase change of water from land to air, is the second
critical component of the hydrological cycle (Zhang et al., 2016; Cui et al., 2021a; Yang et al., 2023; Tang et al., 2024). It accounts for more than 60% of the land surface water that comes from precipitation and returns to the atmosphere (Oki and Kanae, 2006). Therefore, it is essential to accurately estimate the amount and variation of global ET. Ground-based instruments for observing ET are widely distributed globally. The eddy covariance technique is the most commonly used, providing high-frequency (10–20 Hz) measurements of vertical wind speed and water vapor density (Aubinet et al., 2012;
Pastorello et al., 2020). By calculating their covariance, the latent heat flux (LE, equivalent to ET in $W/m^2$; Hereafter LE was used when describing ground observations) is derived. Its advantages are the non-destructive measurement of the underlying surface environment and flexible installation (in contrast to lysimeters) (Baldocchi, 2020; Pastorello et al., 2020). However, challenges remain in practical applications when LE obtained from the eddy covariance technique ($LE_{EC}$) primarily serves the two research communities:

(1) The global change analysis research community. With the abundance of data and the development of models, more and more ET products based on remote sensing or earth system model simulation are produced and shared (Mu et al., 2011; Martens et al., 2017; Zhang et al., 2019; Cui and Jia, 2021; Zheng et al., 2022). However, their results differ significantly in average annual totals, temporal trends and spatial distribution, which prevents us from properly understanding current changes in ET and the water-carbon cycle (Chen et al., 2014; Hu et al., 2021; Cui et al., 2023; Yang et al., 2023; Tang et al.,
2024). Since $LE_{EC}$ data are considered the ground truth, researchers are eager to find evidence from ground observations to support their hypotheses. As the most widely used $LE_{EC}$ dataset, the FLUXNET2015 dataset only provides observations up to 2014 (Pastorello et al., 2020). It cannot support global climate change analysis, nor can it help resolve discrepancies between different products;

(2) ET modelling community. First of all, many ET models (such as PML-V2 and ETMonitor) require $LE_{EC}$ data for
parameter calibration to improve their performance (Zhang et al., 2019; Zheng et al., 2022). Second, all ET products must undergo validation by comparing themselves to $LE_{EC}$ data (Mu et al., 2011; Zhang et al., 2016; Zhang et al., 2019; Cui et al., 2021b; Zheng et al., 2022). Especially for the latest models developed using new satellite data (such as SMAP launched in 2015), there is a need to develop and validate them based on the latest ground-based benchmark data (Das et al., 2018; Zhang et al., 2024). However, due to limitations such as data sharing policies, the research community still relies on
FLUXNET2015 as the primary source for calibration and validation. With the acceleration of the global water and energy cycle, parameters calibrated using outdated data may no longer be applicable today, and it is difficult to assess model performance over the past decade. The research community hopes to use the latest, long-term $LE_{EC}$ data, but there are currently no up-to-date datasets readily available for them.

Therefore, the two main issues with $LE_{EC}$ data, such as those represented by FLUXNET2015, are:

(1) Extensive data gaps. There is a substantial amount of missing data in $LE_{EC}$. The missing rate of hourly data is around 40% and can be up to 70% for some sites. Long gaps, such as the 30-day gap scenario, account for an average of 44% of all missing data in FLUXNET2015. Although the marginal distribution sampling (MDS) algorithm is used as the official gap-filling algorithm, its performance in filling these long gaps is suboptimal (Foltýnová et al., 2020; Zhu et al., 2022);

(2) Limited observation duration. Only 33% of the sites have observation periods exceeding 10 years and few sites have

more than 20 years of observations. After quality control, less than half of the sites have observation periods of more than 8 years. MDS can only be used as a gap-filling algorithm, but not for data prolongation. The potential of $LE_{EC}$ data is not fully exploited. Therefore, there is an urgent need for a long-term ET benchmark dataset based on ground observations with temporal continuous and high-quality data.

To address this, we developed a gap-filling and prolongation framework for $LE_{EC}$ data and a benchmark dataset for global

ET estimation from 2000 to 2022 across 64 sites is established. We selected 64 sites of a total 206 public sites from FLUXNET2015 based on a rigorous filtering criteria. The data obtained from reanalysis and remote sensing data were pre-processed to match the point scale. Then, A novel bias-corrected random forest (RF) algorithm was used as the gap-filling and prolongation algorithm of the framework to produce seamless half-hourly and daily LE data. We designed comprehensively experiments to evaluate our results, including performance under different gap-length scenarios for gap-

filling results, the consistency between forward and backward prolongation, and the temporal stability of the prolongation. This dataset aims to provide valuable data support for global ET modelling, water-carbon cycle monitoring and climate change analysis.

## 2 Data

### 2.1 FLUXNET2015

The FLUXNET2015 dataset contains land-atmosphere exchanges of energy and carbon data from 212 global distributed sites (206 sites under CC-BY 4.0 license) (https://fluxnet.org/data/fluxnet2015-dataset/). We mainly used the LE data observed by eddy covariance technique and some auxiliary meteorological observations. From the original measurements to the hourly / half-hourly product, both data have been through a strict and uniform processing procedure across all sites, and undergone further scrutiny for these critical variables (Pastorello et al., 2020). After quality assurance and quality control,

data that didn't meet the standard or missed due to power failures or sensor malfunction were filtered out and quality control flags were given. Only data marked as 0 is regarded as the ground observations, while others were gap-filled by MDS algorithm with high to low confidence if the number grew larger. We only used the $LE_{EC}$ data and the meteorological data marked as 0.



## 2.2 ERA5-Land

We used the latest Reanalysis v5 dataset (ERA5-Land) provided by the European Centre for Medium-Range Weather Forecasts (ECMWF) (Muñoz-Sabater et al., 2021) as the source of reference data (https://www.ecmwf.int/en/era5-land). It provides global seamless meteorological data at the spatio-temprol resolution of 0.1°× 0.1° and 1 hour since 1950. The dataset provided meteorological variables including air temperature (TA), u-component of wind, v-component of wind, dewpoint temperature, incoming shortwave radiation (SW_IN), incoming longwave radiatio (LW_IN), and air pressure

(PA). Wind speed (WS) was caculated by its two component, and relative humidity (RH) was caculated by the following equations:

$$RH = \frac{e}{e_s} \times 100\% \, , \tag{1}$$

$$e_s = 6.1078 \times \exp\left(\frac{at_a}{t_a+273.15-b}\right) \begin{cases} a = 17.27, \ b = 35.86, t_a > 0 \\ a = 21.87, \ b = 7.66, \ \ t_a \leq 0 \end{cases} , \tag{2}$$

$$e = 6.1078 \times \exp\left(\frac{at_d}{t_d+273.15-b}\right) \begin{cases} a = 17.27, \ b = 35.86, t_a > 0 \\ a = 21.87, \ b = 7.66, \ \ t_a \leq 0 \end{cases} , \tag{3}$$

where $e_s$ is the saturated vapour pressure (kPa), e is the actual vapour pressure (kPa), $t_a$ is the TA (℃) and $t_d$ is the dewpoint temperature (℃).

## 2.3 MODIS

We obtained the remotely-sensed normalized difference vegetation index (NDVI) data derived from Moderate Resolution Imaging Spectroradiometer (MODIS) MYD13Q1.061 dataset. Its spatial resolution is 250 m and data are provided every 16

days. This dataset has been proved to be one of the best NDVI datasets and widely used in ET modelling.

## 3 Methodology

The gap-filling and prolongation framework for LE$_{EC}$ data mainly contained 3 parts: site selection and data pre-processing, gap-filled half-hourly or hourly LE data generation, and prolonged daily LE data generation (Fig 1). The details are shown as follows:



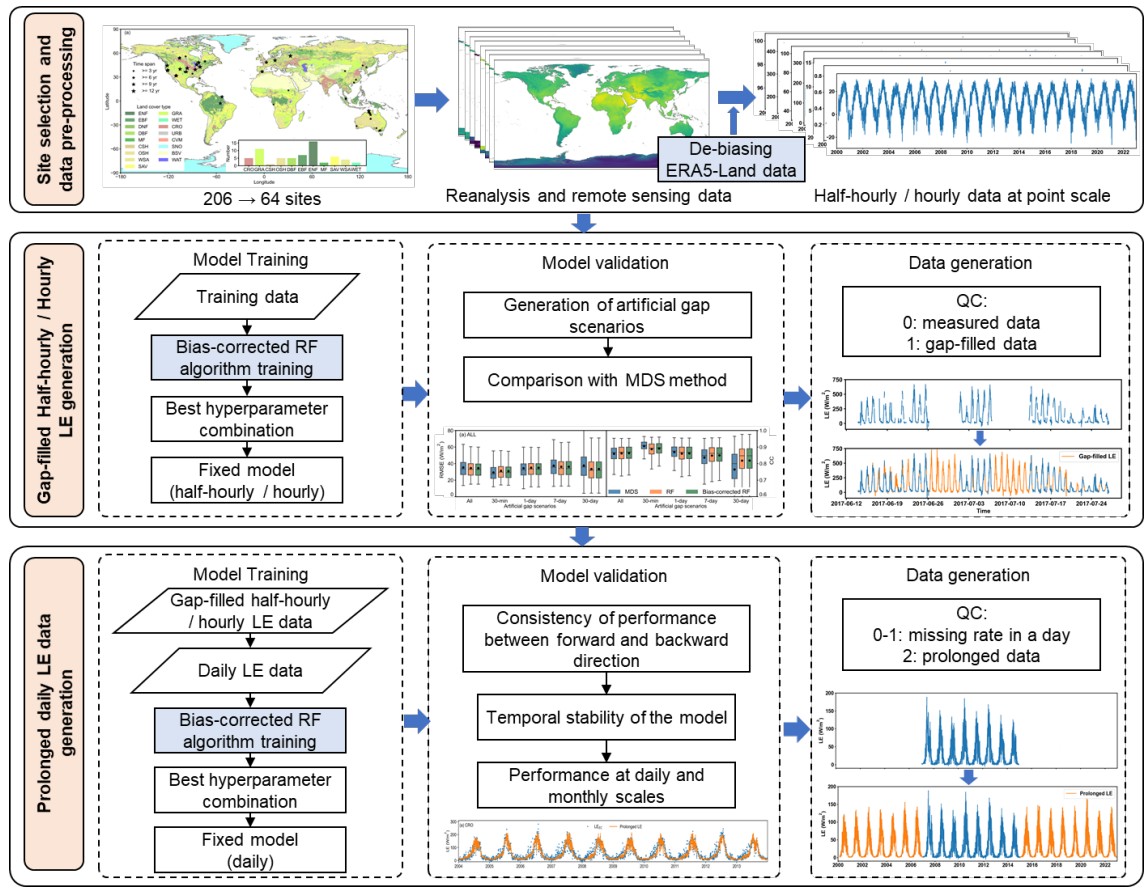

**Figure 1** Schematic of the gap-filling and prolongation framework for $LE_{EC}$ data.

## 3.1 Site selection and data pre-processing

### 3.1.1 FLUXNET2015 site selection

We selected 64 sites from the a total of 206 open-access FLUXNET2015 sites. The filtering criteria included: 1) Time span.
Selected sites must have observations greater than or equal to 3 years. It essential for data prolonging to have sufficient data;
2) data missing rate. The missing rate of the selected sites must under 50%, so that there are enough information for half-
hourly or hourly gap-filling; 3) energy balance closure. We calculated the daily energy balance ratio (EBR) if there were
more than 36 (18 for hourly data) valid observations in a day. The closer the EBR is to 1, the more the observed surface
energy data aligns with the first law of thermodynamics, indicating higher data quality. The site was retained when the
number of days with an EBR of 0.8 to 1.2 accounted for more than 20% of all observed days. The EBR was calculated as
follows:

$$EBR = \frac{\sum_{i=1}^{n}(LE+H)}{\sum_{i=1}^{n}(R_n-G)},$$ (4)

where: $R_n$, G, and H are the net radiation, soil heat flux, and sensible heat flux, respectively.

Specially, there is no eligible sites in Africa strictly based on these criteria, thus we chose 2 more sites with relatively good
data quality. In general, 64 sites were selected (Fig 2). They cover most regions or countries, with 49 sites in north
hemisphere, and 15 sites in south hemisphere. Sites in Europe and the Americas have long observation periods, while Asia
and Oceania are shorter. The average number of years of observations at the site location is around 8 years. Approximately
10-40 sites per year are able to provide observations between 2000 and 2014. Moreover, they represent the vast majority of
vegetated landcover type. See Table A1 for more specific site information.

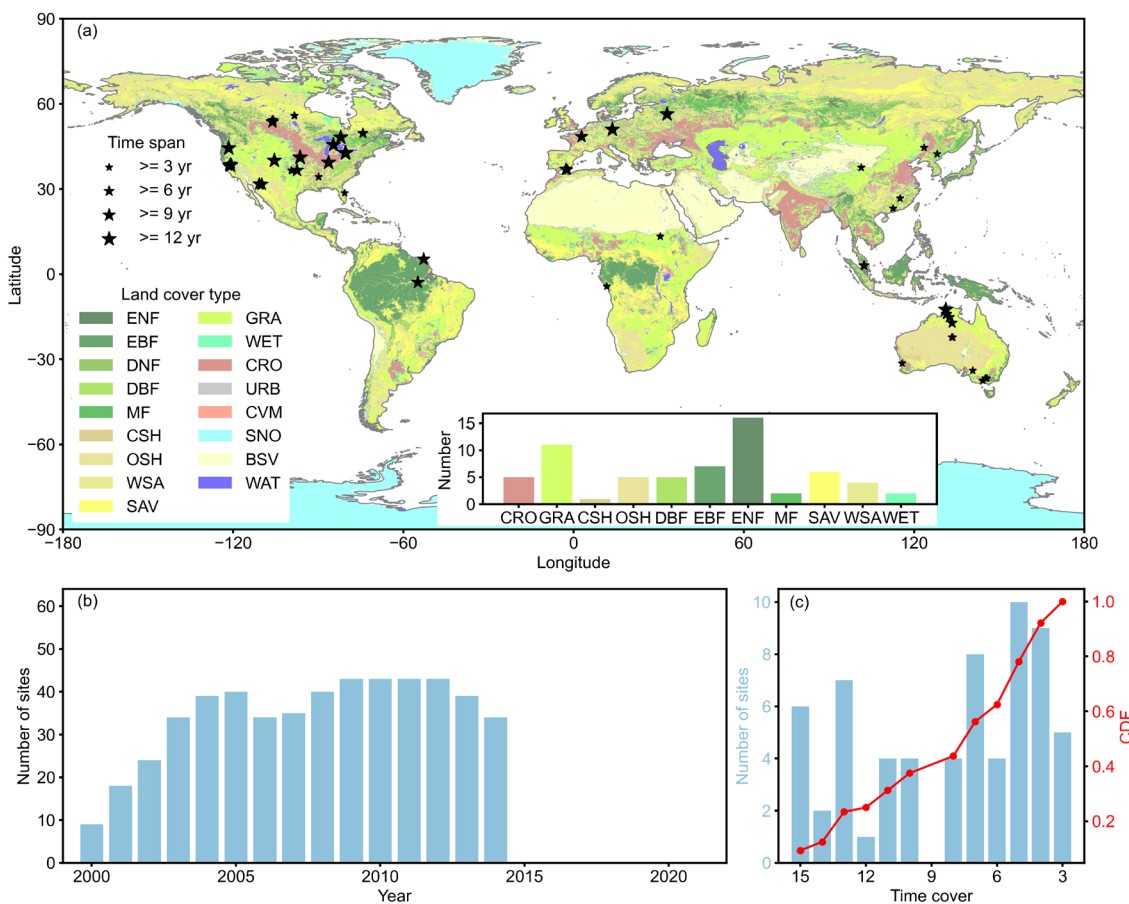


**Figure 2** Global distribution (a) and information (b and c) of 64 selected FLUXNET2015 sites. The size of the star mark
indicates the length of the data record. Panel (b) shows the number of sites in a year from 2000 to 2022. Panel (c) is the
statistic of the length of observation periods for all sites.

### 3.1.2 Data pre-processing

We followd the same data pre-processing procedure by Li et al (2024). For the $LE_{EC}$ data, non-observed were filtered out
based on quality control flags. The rest were the data for training and test datasets. The $LE_{EC}$ data is reported at local time.



Reference variables, including TA, WS, RH, PA, SW_IN, LW_IN, and NDVI, were selected based on Penman-Monteith (PM) equation (Monteith, 1965). These variables directly or indirectly influnce the parameters in PM equation and are the most suitable variables to characterize meteorological and vegetated conditions which influence the ET process (Zhang et al., 2008; Mu et al., 2011; Li et al., 2024a). This equation is expressed as:

$$LE = \frac{\Delta R_n{}^* + \frac{\rho c_p \times VPD}{r_a}}{\Delta + \gamma \left(1 + \frac{r_s}{r_a}\right)} ,$$ (5)

$$VPD = e_s - e ,$$ (6)

where $\Delta$ *is* the slope of the vapor pressure curve, $R_n{}^*$ is the net available radiation at the evaporating surface, $\rho$ is the density of air, $c_p$ is the specific heat of air at constant pressure, *VPD* is the air vapour pressure deficit, $\gamma$ is a psychrometric constant, $r_s$ is the surface resistance, and $r_a$ is the aerodynamic resistance.

Reference variables from ERA5-Land and MODIS were extracted as time-series data at point scale using Google Earth Engine (https://code.earthengine.google.com/). Depending on the frequency of $LE_{EC}$ data records, the hourly time-series data from ERA5-Land were resampled to a half-hourly scale using the linear interpolation method or remain constant at hourly scale. We converted the UTC time to the local time of the site. The NDVI data with a 16-day temporal scale were resampled to a daily frequency using Savitzky-Golay filtering. The same value was then assigned uniformly for each day.

### 3.1.3 De-biasing the ERA5-Land data

In order to minimize mismatch between in-situ and raster data, the time-series data from ERA5-Land were further processed. We followed the similar procedure as the official products (Vuichard and Papale, 2015) and corrected the bias between ground observations and ERA5-Land using linear correction method:

$$Ground_i = k_i \times EL5_i + b_i ,$$ (7)

where: *i* means different variables, EL5 is the ERA5-Land data, and *Ground* is the ground observations from FLUXNET2015. These variables were filtered by quality control flags and only valid observations were used. The ground observed vapour variable was VPD instead of RH in some sites. We transferred it to RH using the following equation:

$$RH = \left(1 - \frac{VPD}{e_s}\right) \times 100\% ,$$ (8)

### 3.2 Gap-filled half-hourly or hourly LE data generation

### 3.2.1 Bias-corrected random forest algorithm

Random Forest (RF), used for both classification and regression tasks, is composed of multiple decision trees, and it combines their predictions to generate the final output (Breiman, 2001). Numerous studies have demonstrated the effectiveness of machine learning algorithms for gap-filling ground-based ET data (Moffat et al., 2007; Irvin et al., 2021; Mahabbati et al., 2021; Zhu et al., 2022; Li et al., 2024a). The random forest (RF) algorithm is considered as the most robust and efficient machine learning algorithm to replace traditional MDS algorithm and has significant potential for prolonging time series. However, there has been a lack of research on prolonging $LE_{EC}$ time series using the RF and no corresponding



datasets have been released. Although performance of RF on flux gap-filling has been proven to be effective, it still faces challenges, such as overestimating lower values and underestimating higher values. Therefore, it is necessary to correct the
bias. Here, we chose a novel bias-corrected RF algorithm (Zhang and Lu, 2012). It added a bias correction RF model to improve the performance compared with original RF (Fig 3). This algorithm has been used for studies such as drought monitoring (Feng et al., 2019; Wang et al., 2023). Here, the bias-corrected RF model for processing flux data were built and the detailed procedure of this bias-correction method is summarized in Fig 3.

In the model training steps, we trained one model (including RF model 1 and 2) for each site, meaning a total of 64 models
were trained for data gap-filling task. The data for each site were randomly divided into two parts. The training dataset was 80% of the total dataset and the rest was the test set. We used a 10-fold cross validation method to determine the optimal combination of hyperparameters to ensure the best model performance and avoid overfitting. Futhurmore, the taining and test dataset were generated 20 times for one site and found that hyperparameters gained from different taining data are strikingly similar for one site. Therefore, for the final data generation, we chose the hyperparameters with reletivly better
performance and used all valid LE observations to build the model. See 3.2.2 for details on how to split the training and test sets.

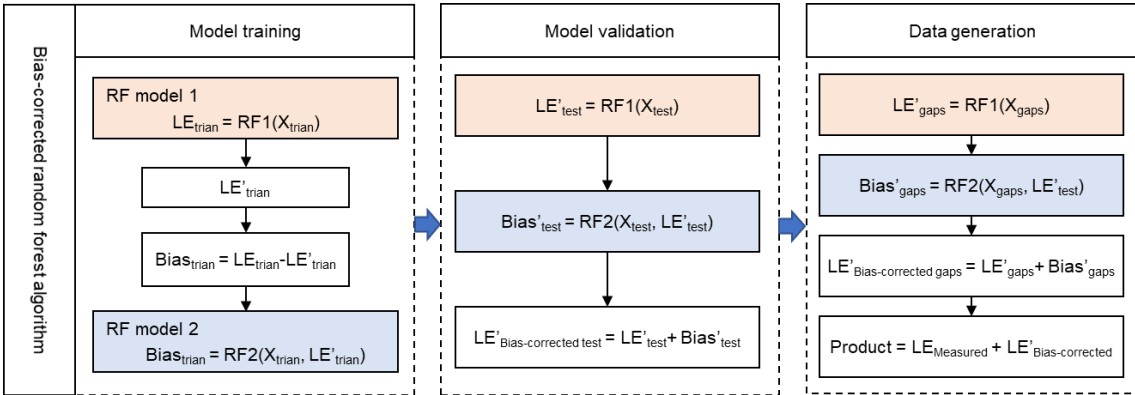

**Figure 3** Schematic diagram of the bias-corrected RF algorithm. Train in the subscript indicates the training data. Test in the subscript indicates the test data. Gaps in the subscript indicates the data gap to be filled. Single quotes indicate predicted
values. X indicates the reference variables, including TA, WS, RH, PA, SW_IN, LW_IN, and NDVI. Prolonging daily data also has the same processing steps.

### 3.2.2 Artificial gap scenarios

The length of gaps in $LE_{EC}$ data varied a lot, from one single missing record to more than 30-day missing data. In order to fully evaluate the performance of our model, we generated four different gap-length scenarios from short to long: 30-min, 1-
day, 7-day and 30-day scenarios (Zhu et al., 2022; Li et al., 2024a). The amount of data for each of the four gap scenarios accounts for about 5% of the total dataset, and all artificial gaps constituted the test set (20%). After removing these artificial gaps, we used the rest data (80%) to train our model.

Specifically, we used a sliding window approach to generate gap scenarios. If the valid observed data coverage within this window exceeded 50%, the window was marked. The window would automatically move forward until this criterion was met and no overlaps among marked sliding windows were ensued. The sliding window size initially started at 30 days. After completing one full round of marking the data, we randomly selected data gaps that account for 5% of the total dataset. and these data were removed from all data. Then, the sliding window size was reduced to 7 days and 1 day, and the steps were repeated. Finally, we randomly removed 5% of half-hourly data to make the 30-min scenarios, ensuring 5 consecutive valid points before and after each gap. To ensure the robustness of the results, we repeated the above steps 20 times, generating 20 different combinations of training and testing sets.

For intercomparison, we also used MDS algorithm and the original RF algorithm as the gap-filling algorithm of the framework to fill the gaps. The core of the MDS algorithm is to use a sliding window approach to find similar meteorological conditions (Reichstein et al., 2005). It primarily uses SW_IN, VPD, and TA as reference variables. The larger the sliding window, the lower the confidence in the gap-filling results. To closely simulate the official data producing process, this study set the minimum thresholds for the three variables at 50 W/m$^2$, 5 hPa, and 2.5°C, respectively. The MDS algorithm was implemented using the REddyProc (R package, v.1.3.3).

### 3.3 Prolonged daily LE data generation

### 3.3.1 Data generation

After half-hourly or hourly gap-filling, the time continues half-hourly data were obtained. We aggregated the continuous LE data and reference data from half-hourly or hourly scales to the daily scale and provided the daily missing rate as the quality control flag. We also chose the bias-corrected RF algorithm as the prolongation algorithm for the framework. The model structure and training steps were the same as in section 3.2.1. During model training, we trained one model for each station, selecting all data except for those with a missing rate of 1 (completely missing) for model training. The 10-fold cross-validation method was used to determine the optimal hyperparameters. Ultimately, the seamless daily LE data from 2000 to 2022 were produced. The final product has been deposited at https://doi.org/10.5281/zenodo.13853409 (Li et al., 2024b) and can be downloaded publicly.

### 3.3.2 Experimental design for evaluating the prolonged data

Since the number of days with a missing rate of 0 at the daily scale is very rare, we consider that a missing rate of less than 10% at the daily scale can be used as the test data. However, using this criterion results in an insufficient training dataset. Therefore, we chose data with daily missing rate of less than 1 as the training set.

The prolongation at the daily scale is divided into two directions: forward and backward. To demonstrate the consistency of our method in both directions, we used the first 1/3 of the data as the test set and the remaining 2/3 of the data as the training set for the backward approach, while employing the first 2/3 as the training set and the last 1/3 as the test set for the forward approach. We compared the performance of both directions.



As the prolongation period increases, the temporal stability of the model's performance also needs to be validated. Among the 64 sites, the average observation period is 8 years, with a minimum of 3 years. Therefore, we selected these two representative lengths for our experiments. First, we chose sites with more than 8 years of observations, using the first 8 years of data as the training set and each subsequent year as the test set. Second, for sites with over 3 years of observations, we used the first 3 years as the training set and each subsequent year as the test set.

**3.4 Performance metrics**

We selected three commonly used performance metrics, including the root mean square error (RMSE, W/m$^2$), bias (Bias, W/m$^2$), correlation coefficient (CC), and coefficient of variation (CV). The equations are as follows:

$$\text{RMSE} = \sqrt{\frac{1}{n}\sum_{i=1}^{n}(p_i - o_i)^2} \, , \tag{9}$$

$$\text{Bias} = \frac{1}{n}\sum_{i=1}^{n}(p_i - o_i) \, , \tag{10}$$

$$CC = \frac{\sum_{i=1}^{n}(p_i - \bar{p})(o_i - \bar{o})}{\sqrt{\sum_{i=1}^{n}(p_i - \bar{p})^2 \sum_{i=1}^{n}(o_i - \bar{o})^2}} \, , \tag{11}$$

$$CV = \frac{\sigma}{\mu} \times 100\% \, , \tag{12}$$

where $p_i$ and $o_i$ are the values from prediction and observation, respectively. $\bar{p}$ denotes the mean predicted value. $\sigma$ is the standard deviation of the target data. $\mu$ is the averaged value of the target data.

**4 Results**

**4.1 Evaluation of half-hourly or hourly gap-filled LE data**

**4.1.1 Gap-filling performance under different gap-length scenarios**

We conducted a comprehensive evaluation of the gap-filling performance for three algorithms under artificially constructed gap scenarios, including the official algorithm (MDS), widely-used RF algorithm, and novel bias-corrected RF algorithm. For each station and each combination of training and test set, we calculated the statistical metrics RMSE, CC, and Bias, and

then visualized the results using box plots (Fig 4 and Fig 5).

In general, the results indicate that the gap-filled data obtained using the bias-corrected RF are superior to the official (MDS) algorithm, particularly outperforming it significantly for long gaps. The bias-corrected RF exhibits the best performance (32.84 W/m$^2$ and 0.87 in terms of median RMSE and CC), with median RMSE improvements of 1.78% and 0.69% compared to MDS and RF, respectively. As for the bias metric, Figure 5 shows that as the length of the gap length increases,

the uncertainty increases and the bias-corrected RF provides more robust results.

For short gaps, we find that the performance of the bias-corrected RF is closer to those of the MDS compared to the original RF. Specifically, the MDS performs exceptionally well, with median values of RMSE and CC at 27.29 W/m$^2$ and 0.91,





respectively. The original RF performs the worst, while the bias-corrected RF reduce bias (2.00% in terms of RMSE), making its performance closer to the MDS compared with the original RF. However, as the gap length increases, the
performance of the MDS declines sharply, which is consistent with previous studies (Foltýnová et al., 2020; Zhu et al., 2022; Li et al., 2024a). Under the 30-day gap length scenario, the median RMSE of the MDS (35.91 W/m$^2$) is 11.02% and 11.41% lower than those of RF (31.95 W/m$^2$) and bias-corrected RF (31.81 W/m$^2$), respectively. Due to the use of the sliding window method, MDS encounters significant issues during the early months of observation. Specifically, when data is completely missing for the early months, the results from MDS at the monthly scale are nearly the same value. A detailed
analysis of this issue can be found in Section 5.1.

We further analyzed the gap-filling results across different land cover types. Based on station count, land cover characteristics, and relevant practices from previous studies, we categorized the land surface types into four groups for analysis: CRO, GRA, DBF/EBF/ENF/MF, and CSH/OSH/SAV/WSA/WET. Overall, for all land surface types, the bias-corrected RF performs better than the original RF and provides closer performance to MDS. Specifically, the bias-corrected
RF shows the most significant improvement in CRO, with the median RMSE being 7.05% lower compared to MDS. This indicates that incorporating NDVI as a reference variable can better capture the seasonal dynamics of crops. We also observed that in GRA and CSH/OSH/SAV/WSA/WET, the bias-corrected RF provides results closer to the gap-filling performance of MDS and the MDS performs much better than the original RF. Across different gap length scenarios, the performance is consistent across land cover types: the bias-corrected RF demonstrates close performance to the MDS and the
RF performs worse than MDS for short gap length. For longer gap length, RF and bias-corrected RF significantly outperform the MDS. Considering that in the FLUXNET2015 dataset, long gaps account for 44% of the data, the bias-corrected RF can serve as a more reliable alternative algorithm to the MDS for hourly-scale data gap-filling, yielding more robust results than those produced by the MDS. Overall, the bias-corrected RF algorithm combines the superior performance of the original RF algorithm under long gap length scenario and provides corrections where the original RF underperforms.

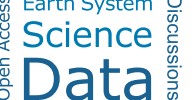




**Figure 4** The gap-filling performance of three algorithms under different gap-length scenarios. The left panels show the results of the root mean square error (RMSE, W/m²) and the right panels show results of correlation coefficient (CC) between gap-filled values and observations. Different rows of this figure indicate different land cover types. The three horizontal lines of the boxes indicate the first quartile, median, and third quartile, respectively, and the black dots indicate

the means. MDS: marginal distribution sampling. RF: random forest.

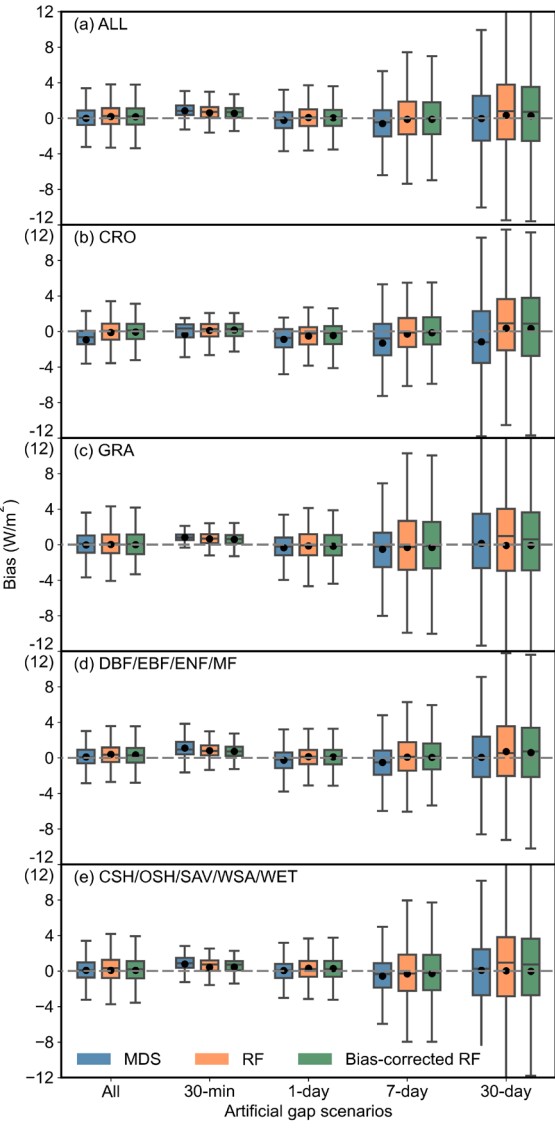

**Figure 5** The bias between gap-filled values and observations of three methods under different gap-length scenarios. Different rows of this figure indicate different land cover types. The three horizontal lines of the boxes indicate the first quartile, median, and third quartile, respectively, and the black dots indicate the means. MDS: marginal distribution sampling. RF: random forest.


### 4.1.2 Examples of gap-filled data under artificial 30-day gap-length scenario

For the 30-day gap scenario, the bias-corrected RF algorithm performs better than MDS in characterizing the time series. As shown in Figure 6, the bias-corrected RF exhibits strong performance across all land cover types and provides a more accurate representation of daily periodic variations. Although there are still some biases in predicting certain extremes, these

are generally smaller compared to those of MDS. In contrast, MDS demonstrates significant gap-filling biases across



different land cover types, resulting in abnormal overestimations and underestimations (Fig 6a, b, and i). In some cases, it even fails to capture the daily variations of LE (Fig 6e), while also distorting irregular LE changes (Fig 6c).





**Figure 6** Time series of gap-filled results obtained from the bias-corrected RF algorithm compared to those from the MDS

algorithm under artificial 30-day gap-length scenario across different land cover types. The blue dashed boxes indicate scenarios where the MDS gap-filling results are significantly biased. The sites corresponding to each land cover type are: US-ARM, CN-Cng, FR-Fon, BR-Sa1, RU-Fyo, CA-Gro, US-KS2, ES-LJu, SD-Dem, AU-How, and US-Myb.

## 4.2 Evaluation of daily prolonged LE

### 4.2.1 The consistency between forward and backward prolongation.

As shown in Fig 7a and 7b, the prolongation performance in both forward and backward directions exhibit high consistency. The results have good accuracy, with RMSE (CC) of 16.58 W/m$^2$ (0.91) for forward and 17.35 W/m$^2$ (0.90) for backward. The slightly difference may be mainly due to a higher volume of missing data in the first two-thirds of the data compared to the last two-thirds for sites of these land cover types (See Section 5.1). There are slight variations in prolongation results for different land cover types (Fig 7c and 7d). Performance of CRO and DBF/EBF/ENF/MF is almost the same in both

directions. Similar to the half-hourly data gap-filling, our results also demonstrate excellent performance in cropland, with a CC of 0.93 in both directions. GRA and CSH/OSH/SAV/WSA/WET perform slightly worse (2.46 W/m$^2$ and 3.74 W/m$^2$ higher) in the backward direction.

Figure 2b indicates that the need for forward prolongation is significantly greater than for backward prolongation from 2000 to 2022. Therefore, the validation in the following sections will focus only on the forward direction.
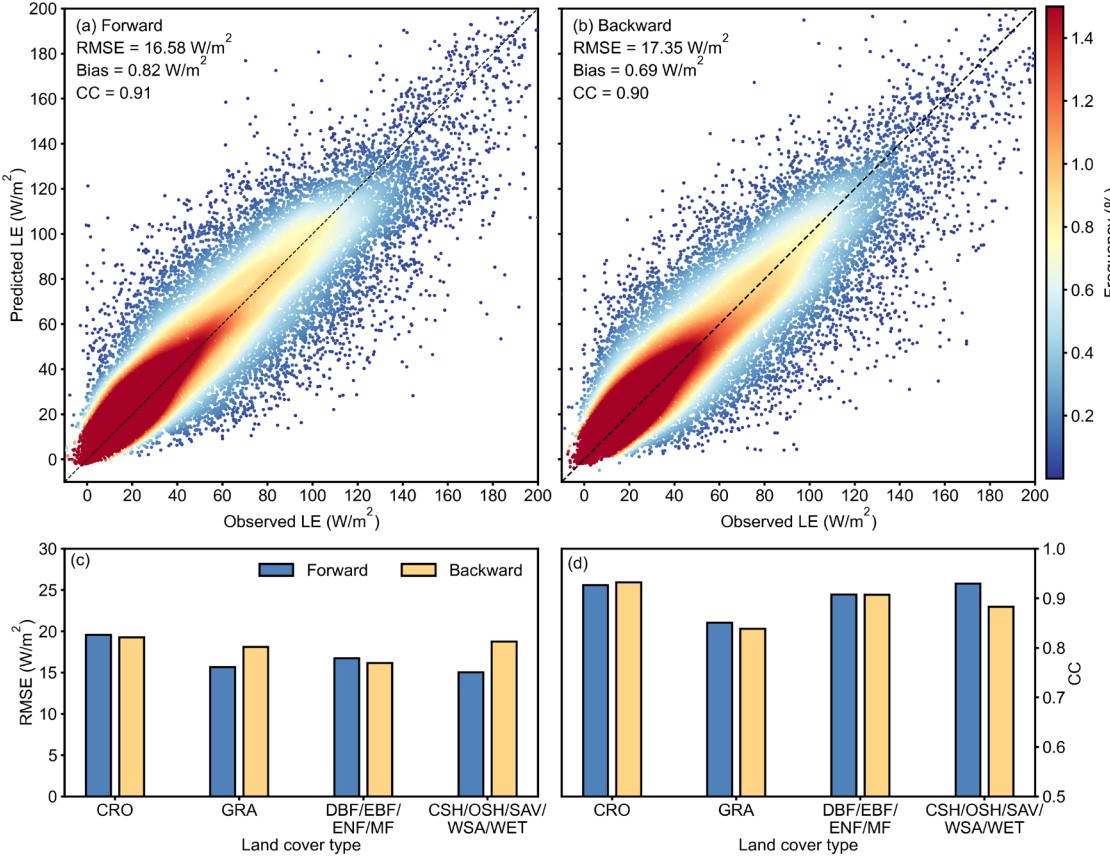

**Figure 7** The consistency of forward and backward prolongation. (a) and (b) show the scatterplots of predicted daily LE against observations for forward and backward prolongation, respectively. (c) and (d) are the specific performance of different land cover types.

### 4.2.2 The temporal stability of the prolongation

We used data from the first three years and the first eight years for training and evaluated the prolongation performance for each subsequent year. Three years of data represents an extreme case of the minimum training data volume in this dataset, while eight years of data reflects a typical scenario within the dataset. Figure 8 shows that our prolongation results exhibits minimal performance degradation over time. The greater the amount of training data, the higher the temporal stability will be. Specifically, the model trained using the first three years yields CVs of RMSE and CC of only 3.29% and 3.83%, respectively. The model trained using the first eight years yields CVs of RMSE and CC of only 3.24% and 1.75%, respectively. The bias fluctuates within a small range around zero each year, indicating that our estimation bias is relatively robust. For different land cover types, DBF/EBF/ENF/MF shows good stability. GRA and CSH/OSH/SAV/WSA/WET show more noticeable fluctuations over time but did not experience significant performance degradation. Overall, our model demonstrates excellent temporal stability in both extreme and typical cases.



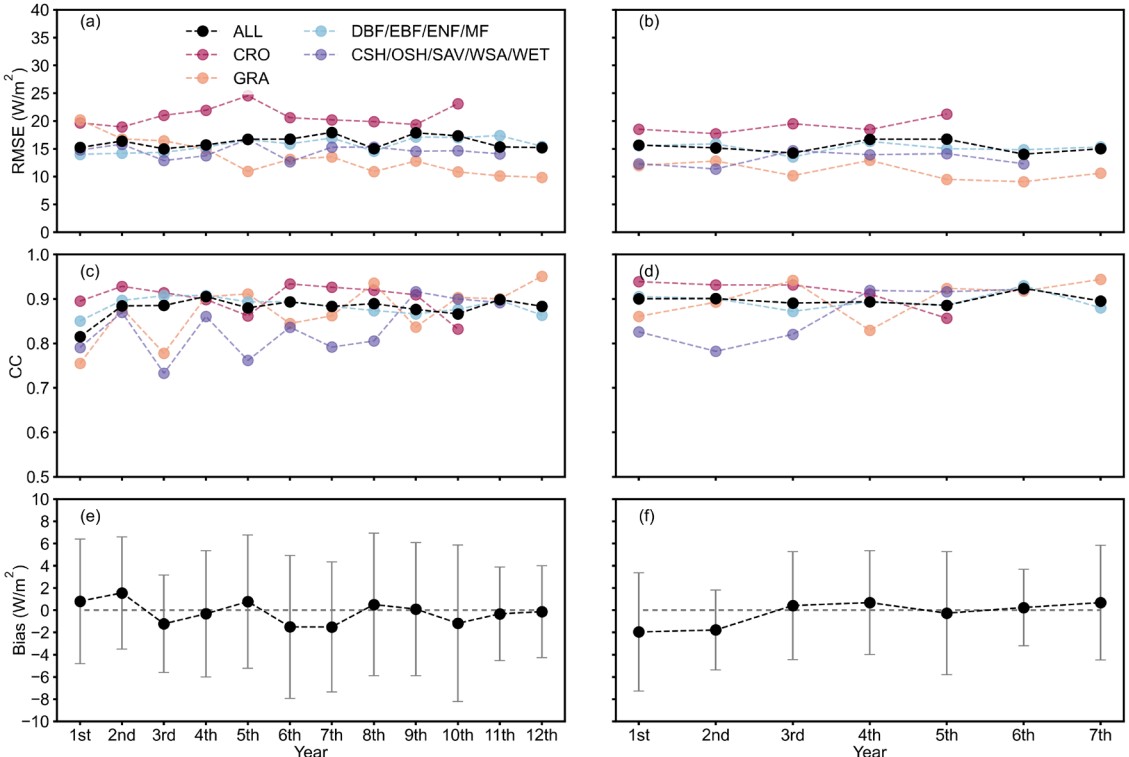

**Figure 8** The temporal stability of the prolongation algorithm for different land cover types. (a), (c), and (e) show the median of RMSE, CC and Bias obtained from the model trained by first 3 years data, respectively. (b), (d), and (f) show the median of RMSE, CC and Bias of obtained from the model trained by first 8 years data, respectively.

**4.2.3 Demonstration of daily- and monthly-scale prolonged time series**

Due to the scarcity of days with a missing rate below 10%, we chose to compare the prolonged results from section 4.2.2 with the daily data aggregated from the hourly gap-filled data. We plotted the results obtained in section 4.2.2 as time series graphs and compared the prolonged results with the aggregated daily data from hourly gap-filled results. As shown in Fig 9 and 10, our prolongation algorithm effectively captures the seasonal variation of LE, aligning well with hourly gap-filled results in both magnitude and trend. The model performs excellently in both extreme (3 years data) and typical (8 years data) cases, particularly for sites with a land cover type of CRO. For evergreen vegetation sites (ENF and EBF) and sparse vegetation sites (SAV and OSH), the lack of vegetation change information leads to unclear influencing factors on LE variation. Some extreme high values are underestimated. However, our algorithm still performs well in capturing daily fluctuations.

Given that many global change studies focus on monthly scales, we aggregated both the daily data to assess the performance. As shown in Fig 11, the monthly scale results meet the requirements of related research. Both the trend and magnitude align well with hourly gap-filled results. The CRO sites match almost perfectly with the hourly gap-filled results, while the ENF and EBF sites, which performed slightly worse at the daily scale, accurately capture subtle fluctuations at the monthly scale.





**Figure 9** Time series of daily prolonged results obtained from the model trained using the first three years across different
land cover types. The sites corresponding to each land cover type are: US-Ne1, AU-DaP, FR-Fon, BR-Sa1, RU-Fyo, CA-Gro, US-KS2, US-Whs, SD-Dem, AU-How, and US-Myb.

**Figure 10** Time series of daily prolonged results obtained from the model trained using the first eight years across different
land cover types. The sites corresponding to each land cover type are: US-Ne1, US-Var, FR-Fon, BR-Sa1, RU-Fyo, CA-Gro,
ES-LJu, and AU-How.

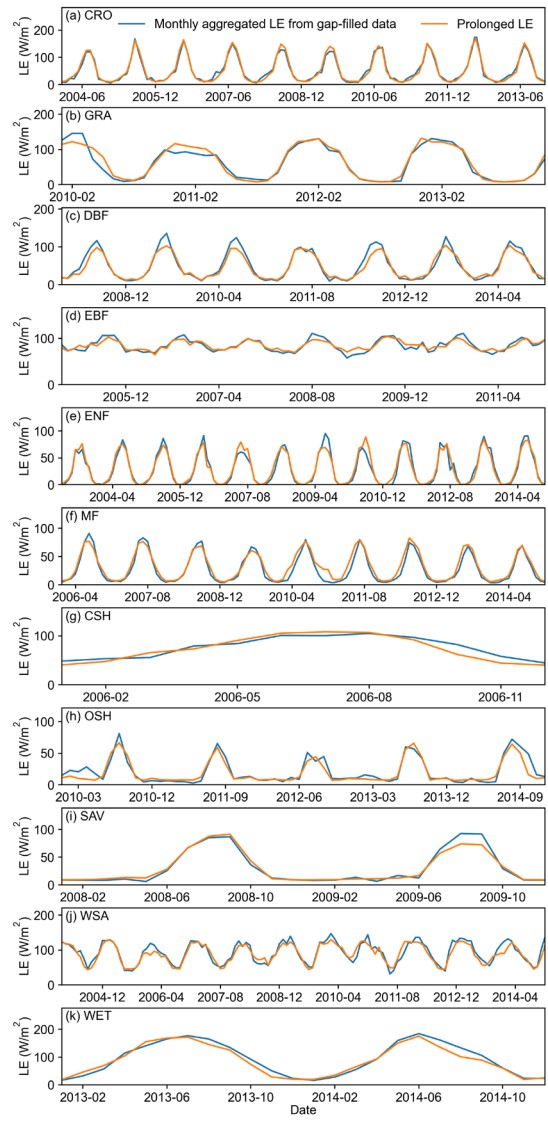

**Figure 11** Time series of monthly aggregated results obtained from the model trained using the first three years across different land cover types. The stations corresponding to each land cover type are: US-Ne1, AU-DaP, FR-Fon, BR-Sa1, RU-Fyo, CA-Gro, US-KS2, US-Whs, SD-Dem, AU-How, and US-Myb.



## 5 Discussions

### 5.1 Comparison between FLUXNET2015 and our dataset

After extensive analysis of the experimental design results in Section 4, we have demonstrated excellent gap-filling and prolongation performance at the methodological level. To evaluate our released dataset, we compared it with the official dataset from FLUXNET2015. This is because missing data in observations cannot provide a verifiable truth. Figure 12 shows the data distribution results of gap-filled data at both hourly and daily scales for the two datasets. The results indicate a high consistency in data distribution between our dataset and FLUXNET2015. At the hourly scale, the median and quartiles of both datasets are nearly identical. For CRO, FLUXNET2015 exhibits slightly higher values compared to our dataset, while for GRA and CSH/OSH/SAV/WSA/WET, its estimates are slightly lower. At the daily scale, the consistency is even greater, with almost identical data distributions across all land surface types.

Additionally, we compared the differences between the two datasets aggregated to monthly and yearly scales. As shown in Fig 13, the data from both datasets distributes along the 1:1 line at both monthly and yearly scales. Although some months and years exhibited discrepancies between the two datasets, it still demonstrates a high degree of consistency. Specifically, at the monthly scale, we observed instances where some LE data of FLUXNET2015 show close values, while our predictions demonstrate clear distinctions. When aggregated to the yearly scale, these discrepancies manifested as outliers. This instance arises because many FLUXNET2015 sites experienced complete data loss for the first four to eight months (e.g., AU-ASM from January to August 2010, CA-Cro from January to July 2003, US-UMd from January to April 2007, among others). Due to the lack of neighboring information in the sliding window, the MDS algorithm struggled to provide effective gap-filling, resulting in nearly identical gap-filled values for those months. Consequently, these months could not be included in the usable data range, rendering the aggregated results at the yearly scale unreliable. In contrast, our algorithm can utilize the reference data for each specific moment to predict the corresponding LE, so we can provide more accurate gap-filling results.

Therefore, the advantages of our dataset are: 1) Hourly scale Gap-filling enhances accuracy compared to FLUXNET2015 under long gap-length scenarios; 2) daily scale results show good consistency with FLUXNET2015 while providing a much longer time series (23 years compared to averaged 8 years). However, our data does have some limitations. For instance, due to the restrictions of NDVI data, our dataset only provides data from February 18, 2000 for both hourly gap-filling and daily prolongation.

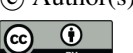



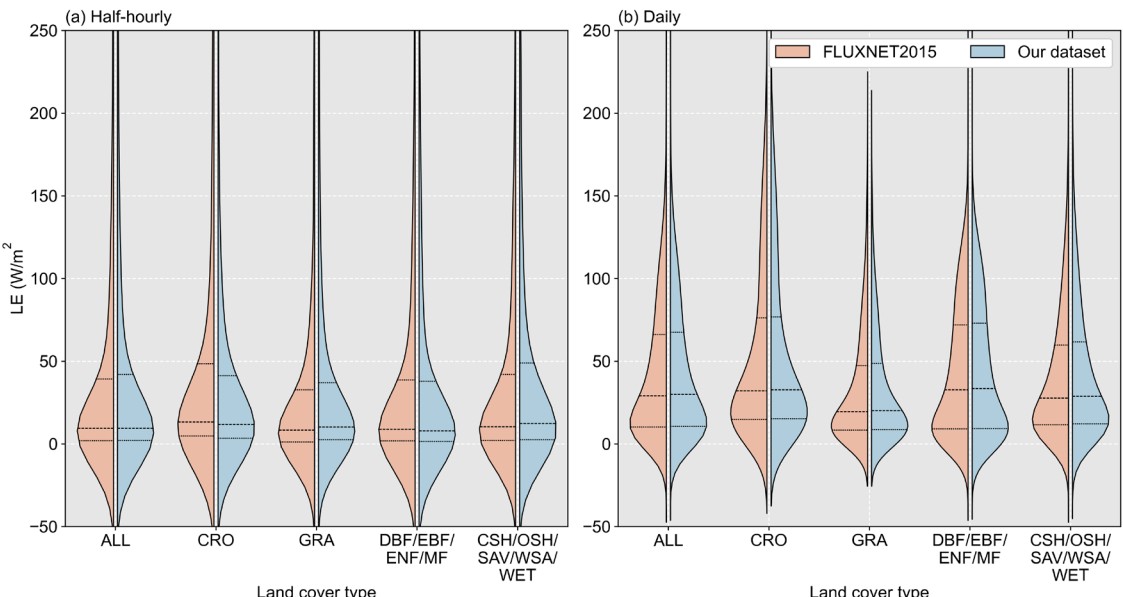

**Figure 12** Distribution of gap-filled data at both (a) half-hourly and (b) daily scales for our dataset and FLUXNET2015 dataset.

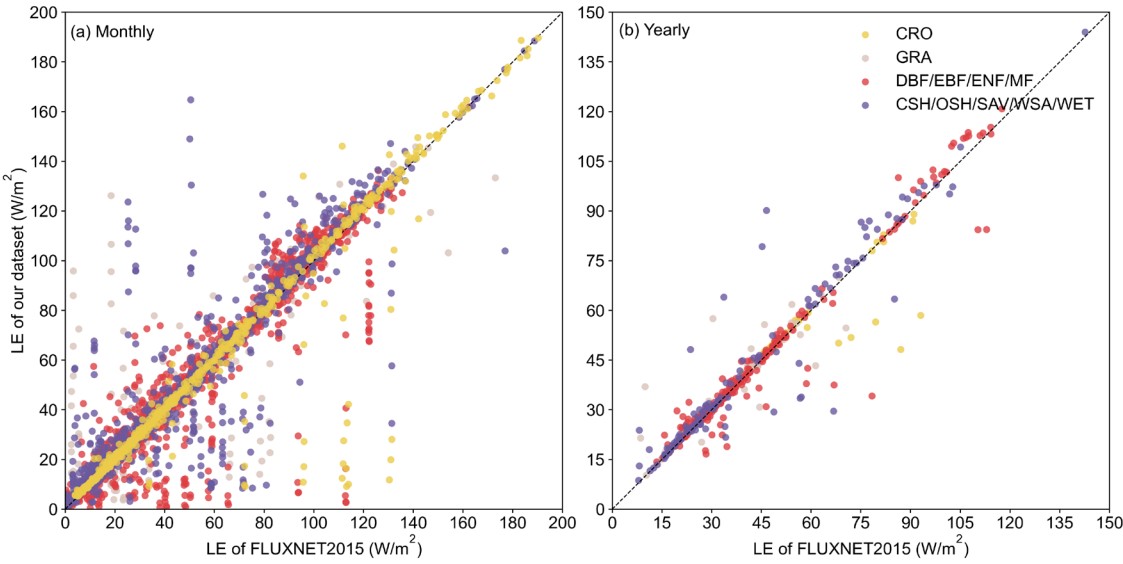


**Figure 13** Scatterplot of LE data of our dataset against that of FLUXNET2015 dataset.

## 5.2 Reference variables importance analysis

Figure 14 presents the results of the reference variables importance using the permutation feature importance technique. Each input feature is randomly shuffled to calculate the performance deterioration. For half-hourly or hourly gap-filling, the





order of variable importance is SW_IN > NDVI > TA > LW_IN > RH > WS > PA. Consistent with earlier research (Irvin et al., 2021; Zhu et al., 2022; Li et al., 2024a), SW_IN is the key variable that significantly influences LE variations across terrestrial ecosystems. It provides energy for the ET process. Throughout the day, SW_IN exhibits significant diurnal variation. NDVI is the second most important variable, but its influence varies between sites. This explains why the performance of the two land cover types in section 4.2.3 is slightly inferior to that of other types. For sites with evergreen

vegetation, seasonal changes in vegetation are not pronounced, making NDVI less effective in providing clear information to the model. For daily prolongation, the order of variable importance is different. The importance of SW_IN decreases significantly because daily LE variation is more closely related to NDVI, which reflects seasonal changes. Similar to the hourly scale, NDVI also shows inconsistencies between sites for the same reasons. Additionally, TA, as the third most important variable, provides critical information at sites dominated by soil evaporation. Variables like LW_IN, RH, WS, and

PA hold comparable significance as minor factors, offering insights into the meteorological background conditions.

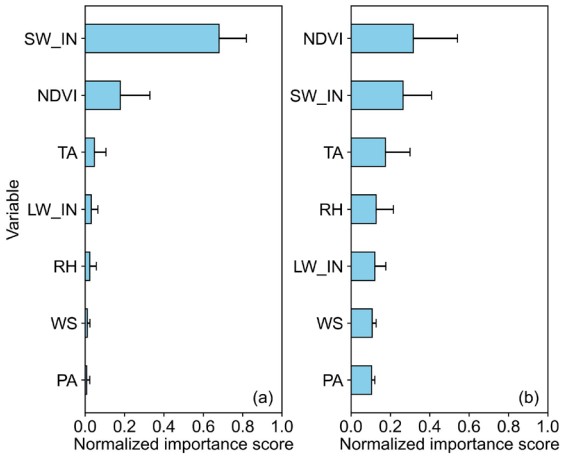

**Figure 14** Variable importance for half-hourly or hourly gap-filling and daily prolongation.

## 6 Data availability

Our released data mainly contains four types of data:

(1) Half-hourly or hourly gap-filled LE data: The data are well gap-filled LE data using the novel bias-corrected RF algorithm. In the filenames, "HH" or "HR" indicate half-hourly or hourly scale data, respectively. The time information in the data files includes a pair of timestamps consistent with those in FLUXNET2015. The data are recorded at local time. The start time is "2000-02-18, 00:00:00", and the end time is the same as the observation time at each site. For the quality control flags (QC), a value of 0 indicates observed data, while 1 indicates gap-filled data.





(2) Prolonged daily LE data: This dataset provides the prolonged daily LE data using the novel bias-corrected RF algorithm. The seamless data covers the period from February 18, 2000, to December 31, 2022. For the prolonged part, the quality flag is set to 2. The rest data is consistent with the aggregated daily LE data.

    (3) Aggregated daily, monthly and yearly LE data: The hourly dataset is aggregated from the gap-filled half-hourly data to a daily scale. The start time is "2000-02-18", and the end time is the same as the observation time at each site. Data quality

control flags are also provided, with the values representing the percentage of hourly observations for each day. The monthly and yearly LE data are aggregated from the prolonged daily LE data. Quality control flags represent the proportion of days with more than 90% of hourly observations in a given month or a given year. No distinction is made between prolonged data and data with complete missing observations within a day. The start time for the monthly data is March 2000, and that for the yearly data is 2001.

All files are formatted as csv files. NDVI and debiased reference variables from ERA5-Land are also provided in our released data. The product has been deposited at https://doi.org/10.5281/zenodo.13853409 (Li et al., 2024b) and can be downloaded publicly.

**7 Conclusions**

    The current $LE_{EC}$ data are increasingly insufficient to meet the needs of two major research communities for for long time-

series benchmark data to support climate change analysis, construction of new models, and validation of new products. To address the issues in the FLUXNET2015 dataset and meet the corresponding requirements, we developed a gap-filling and prolongation framework for $LE_{EC}$ data and a benchmark dataset for ground-based ET from 2000 to 2022 across 64 sites is established. The results indicate that:

    1) Hourly gap-filling: the novel bias-corrected RF algorithm demonstrates excellent performance, achieving a median RMSE

of 32.84 $W/m^2$. It improves the original RF algorithm's poor gap-filling performance for short gaps, approaching the performance of official algorithm (MDS). It also significantly improves performance for long gaps, exceeding the MDS algorithm by 11.41%. The extreme values are predicted more accurately, which reduces the result's uncertainty compared to the MDS algorithm. It performs well across various land surface types, with the most significant improvement (7.05%) observed in cropland. This indicates that including NDVI in reference variables better captures the seasonal dynamics of LE.

Furthermore, our gap-filled data distribution aligns well with official products.

    2) Daily prolongation: our method exhibits robust performance in both forward and backward directions (16.58 $W/m^2$ and 17.35 $W/m^2$, respectively). The method shows slight variation in performance across different land surface types, with the best performance for cropland. In terms of the temporal stability, our results maintain excellent performance under both extreme condition (training with the first three years of data) and typical condition (training with the first eight years of

data). The time series effectively captures seasonal variations in LE, aligning well with observations.



3) For hourly data gap-filling, SW_IN is the most important factor, while NDVI plays a decisive role for daily prolongation. However, in cases where the land surface is dominated by evergreen or sparse vegetation, the importance of NDVI significantly decreases.

Overall, our proposed gap-filling and prolongation framework for LE$_{EC}$ data is robust and a benchmark dataset for global ET
estimation based on FLUXNET2015 from 2000 to 2022 is established. It can provide essential data support for ET modelling, water-carbon cycle monitoring and climate change analysis.

**Appendix A: Site information**

| Site | IGBP | Latitude | Longitude | Start year | End year | Time cover after 2000 | Missing ratio |
|------|------|----------|-----------|------------|----------|-----------------------|---------------|
| AU-ASM | SAV | -22.28 | 133.25 | 2010 | 2014 | 5 | 0.37 |
| AU-Cpr | SAV | -34.00 | 140.59 | 2010 | 2014 | 5 | 0.28 |
| AU-DaP | GRA | -14.06 | 131.32 | 2007 | 2013 | 7 | 0.36 |
| AU-DaS | SAV | -14.16 | 131.39 | 2008 | 2014 | 7 | 0.21 |
| AU-Dry | SAV | -15.26 | 132.37 | 2008 | 2014 | 7 | 0.45 |
| AU-Gin | WSA | -31.38 | 115.71 | 2011 | 2014 | 4 | 0.44 |
| AU-How | WSA | -12.49 | 131.15 | 2001 | 2014 | 14 | 0.35 |
| AU-Rig | GRA | -36.65 | 145.58 | 2011 | 2014 | 4 | 0.26 |
| AU-Stp | GRA | -17.15 | 133.35 | 2008 | 2014 | 7 | 0.29 |
| AU-TTE | GRA | -22.29 | 133.64 | 2012 | 2014 | 3 | 0.40 |
| AU-Whr | EBF | -36.67 | 145.03 | 2011 | 2014 | 4 | 0.32 |
| AU-Wom | EBF | -37.42 | 144.09 | 2010 | 2014 | 5 | 0.41 |
| BR-Sa1 | EBF | -2.86 | -54.96 | 2002 | 2011 | 10 | 0.27 |
| BR-Sa3 | EBF | -3.02 | -54.97 | 2000 | 2004 | 5 | 0.47 |
| CA-Gro | MF | 48.22 | -82.16 | 2003 | 2014 | 12 | 0.24 |
| CA-NS2 | ENF | 55.91 | -98.52 | 2001 | 2005 | 5 | 0.49 |
| CA-NS3 | ENF | 55.91 | -98.38 | 2001 | 2005 | 5 | 0.33 |
| CA-Oas | DBF | 53.63 | -106.20 | 1996 | 2010 | 11 | 0.17 |
| CA-Qfo | ENF | 49.69 | -74.34 | 2003 | 2010 | 8 | 0.23 |
| CA-SF1 | ENF | 54.49 | -105.82 | 2003 | 2006 | 4 | 0.37 |
| CA-SF2 | ENF | 54.25 | -105.88 | 2001 | 2005 | 5 | 0.35 |
| CA-SF3 | OSH | 54.09 | -106.01 | 2001 | 2006 | 6 | 0.36 |
| CA-TP1 | ENF | 42.66 | -80.56 | 2002 | 2014 | 13 | 0.47 |
| CA-TP3 | ENF | 42.71 | -80.35 | 2002 | 2014 | 13 | 0.41 |
| CA-TP4 | ENF | 42.71 | -80.36 | 2002 | 2014 | 13 | 0.18 |
| CG-Tch | SAV | -4.29 | 11.66 | 2006 | 2009 | 4 | 0.57 |
| CN-Cha | MF | 42.40 | 128.10 | 2003 | 2005 | 3 | 0.24 |
| CN-Cng | GRA | 44.59 | 123.51 | 2007 | 2010 | 4 | 0.27 |
| CN-Din | EBF | 23.17 | 112.54 | 2003 | 2005 | 3 | 0.31 |





| | | | | | | | |
|---|---|---|---|---|---|---|---|
| CN-Ha2 | WET | 37.61 | 101.33 | 2003 | 2005 | 3 | 0.17 |
| CN-Qia | ENF | 26.74 | 115.06 | 2003 | 2005 | 3 | 0.21 |
| DE-Obe | ENF | 50.79 | 13.72 | 2008 | 2014 | 7 | 0.18 |
| DE-Tha | ENF | 50.96 | 13.57 | 1996 | 2014 | 15 | 0.13 |
| ES-Amo | OSH | 36.83 | -2.25 | 2007 | 2012 | 6 | 0.38 |
| ES-LJu | OSH | 36.93 | -2.75 | 2004 | 2013 | 10 | 0.27 |
| FR-Fon | DBF | 48.48 | 2.78 | 2005 | 2014 | 10 | 0.18 |
| GF-Guy | EBF | 5.28 | -52.92 | 2004 | 2014 | 11 | 0.24 |
| MY-PSO | EBF | 2.97 | 102.31 | 2003 | 2009 | 7 | 0.21 |
| RU-Fyo | ENF | 56.46 | 32.92 | 1998 | 2014 | 15 | 0.22 |
| SD-Dem | SAV | 13.28 | 30.48 | 2005 | 2009 | 5 | 0.64 |
| US-AR1 | GRA | 36.43 | -99.42 | 2009 | 2012 | 4 | 0.23 |
| US-AR2 | GRA | 36.64 | -99.60 | 2009 | 2012 | 4 | 0.33 |
| US-ARM | CRO | 36.61 | -97.49 | 2003 | 2012 | 10 | 0.15 |
| US-Blo | ENF | 38.90 | -120.63 | 1997 | 2007 | 8 | 0.36 |
| US-Goo | GRA | 34.25 | -89.87 | 2002 | 2006 | 5 | 0.40 |
| US-KS2 | CSH | 28.61 | -80.67 | 2003 | 2006 | 4 | 0.27 |
| US-Me2 | ENF | 44.45 | -121.56 | 2002 | 2014 | 13 | 0.14 |
| US-Me3 | ENF | 44.32 | -121.61 | 2004 | 2009 | 6 | 0.26 |
| US-MMS | DBF | 39.32 | -86.41 | 1999 | 2014 | 15 | 0.34 |
| US-Myb | WET | 38.05 | -121.77 | 2010 | 2014 | 5 | 0.32 |
| US-Ne1 | CRO | 41.17 | -96.48 | 2001 | 2013 | 13 | 0.15 |
| US-Ne2 | CRO | 41.16 | -96.47 | 2001 | 2013 | 13 | 0.23 |
| US-Ne3 | CRO | 41.18 | -96.44 | 2001 | 2013 | 13 | 0.21 |
| US-NR1 | ENF | 40.03 | -105.55 | 1998 | 2014 | 15 | 0.22 |
| US-SRC | OSH | 31.91 | -110.84 | 2008 | 2014 | 7 | 0.38 |
| US-SRG | GRA | 31.79 | -110.83 | 2008 | 2014 | 7 | 0.14 |
| US-SRM | WSA | 31.82 | -110.87 | 2004 | 2014 | 11 | 0.12 |
| US-Ton | WSA | 38.43 | -120.97 | 2001 | 2014 | 14 | 0.31 |
| US-Twt | CRO | 38.11 | -121.65 | 2009 | 2014 | 6 | 0.36 |
| US-UMB | DBF | 45.56 | -84.71 | 2000 | 2014 | 15 | 0.23 |
| US-UMd | DBF | 45.56 | -84.70 | 2007 | 2014 | 8 | 0.17 |
| US-Var | GRA | 38.41 | -120.95 | 2000 | 2014 | 15 | 0.21 |
| US-Whs | OSH | 31.74 | -110.05 | 2007 | 2014 | 8 | 0.16 |
| US-Wkg | GRA | 31.74 | -109.94 | 2004 | 2014 | 11 | 0.16 |

**Author contributions**

Conceptualization: WL and YC. Methodology: WL, ZY, and YC. Data curation: ZY, YC. Funding acquisition: YC. Writing
(initial): WL. Writing (review and editing): ZY, YQ, HY, LS, LW, YS, and YC. Supervision: YC.



## Competing interests

The contact author has declared that none of the authors has any competing interests.

## Acknowledgements

The authors would like to thank the scikit-learn (https://scikit-learn.org/stable/install.html) team and the ReddyProc
(https://cran.r-project.org/web/packages/REddyProc/index.html) team for the packages that help their method establishment
and validation. They also thank the FLUXNET and the research groups for providing the CC-BY-4.0 (Tier one) open-access
eddy covariance data (https://fluxnet.org/data/fluxnet2015-dataset/). They thank the ECWMF team for the public ERA5-
Land reanalysis data (https://www.ecmwf.int/en/era5-land) and the MODIS science team for the MYD13Q1 data.
Additionally, they also thank the Google Earth Engine platform for downloading ERA5-Land and MYD13Q1 data
efficiently.

## Financial support

This study was financially supported by the National Natural Science Foundation of China (Grant No. 42471375 and No.
42130104) and the Key R&D Program of the Ministry of Science and Technology, China (Grant No. 2022YFC3002802).





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
