# Peer review of "A benchmark dataset for global evapotranspiration estimation based on FLUXNET2015 from 2000 to 2022"

_Earth System Science Data, 2024_

## Author Response (AR1)

**Referee 1:**

**General comment:**

The research article discusses the development of a benchmark dataset for global evapotranspiration (ET) estimation, addressing limitations in existing latent heat flux (LE) data from the FLUXNET2015 dataset. Current datasets suffer from short observation periods and significant data gaps, hindering climate change analysis and model validation. To overcome these challenges, the authors created a gap-filling and prolongation framework that generates seamless half-hourly and daily LE data from 2000 to 2022 across 64 sites. They employed a novel bias-corrected random forest algorithm for improved data accuracy, achieving a median RMSE of 32.84 W/m² for hourly and 16.58 W/m² for daily data. The resulting dataset enhances ET modeling, water-carbon cycle monitoring, and climate change research.

The study is one of the pioneering efforts to utilize a bias-corrected random forest approach to enhance data gap-filling performance. I suggest minor revisions to address some specific questions before proceeding with publication.

**Reply:** Thank you for taking your precious time to review our article and also giving us such a positive comment. We have made a detailed reply and they are as follows. We have also carefully examined the entire manuscript and corrected some ambiguous or incorrect expressions therein. All the changes are marked in red.

**Detailed comments:**

**1:** Figure 3 - From the diagram there are two RF models being trained and evaluated. Please indicate that LE and Bias without single quote serve as observational ground-truth labels in Model training box.

**Reply:** Thank you for your suggestion. We have added the explanation of LE and Bias without single quote in the caption of Figure 3.

> **In revised paper Line 194-195:**
> *"LE and Bias with single quotes indicate predicted values, whereas those without single quotes indicate ground observations."*

**2:** In Model validation box, there is only predicted values instead of true values being indicated. Please add that true LE and Bias are used to evaluate the performance of RF1 and RF2 and indicate performance metrics used for each model validation.

**Reply:** Thank you for your suggestion. According to your opinion, we have made corresponding modifications in Figure 3.

Firstly, we have changed the "Model validation" box to the "Model test" box to avoid misunderstanding.

Secondly, for RF1, we added true LE ($LE_{test}$) and used RMSE, CC, Bias as the performance metrics to evaluate the performance of RF1 (the original RF); for RF2, our aim is to accurately calculate and evaluate the predicted LE rather than Bias, so we directly evaluated the performance of RF1+RF2 ($LE'_{Bias\_corrected\ test}$ versus $LE_{test}$) and also used RMSE, CC, Bias as the performance metrics. We have added all the information into the "Model test box".

*In revised paper Figure 3:*

[Figure]

*Figure 3 Schematic diagram of the bias-corrected RF algorithm. Train in the subscript indicates the training data. Test in the subscript indicates the test data. Gaps in the subscript indicates the data gap to be filled. LE and Bias with single quotes indicate predicted values, whereas those without single quotes indicate ground observations. X indicates the reference variables, including TA, WS, RH, PA, SW_IN, LW_IN, and NDVI. Prolonging daily data also has the same processing steps.*

**3:** Figure 4 – It is hard to conclude Bias-corrected RF has better performance than the other two approaches as the mean values of RMSE of those three are tightly close to each other shown in the figure. Consider adding data labels to the mean RMSE values in the figure to highlight the findings. Same for Figure 5.

**Reply:** Thank you for your suggestion. Following your suggestions, we have added data labels of mean RMSE, mean CC, and mean Bias for "All sites" in the first line of Figure 4 and Figure 5. We calculated the average of the four scenarios as the result for the "ALL" scenario and use mean values of RMSE, CC, and Bias to replace the median values. We have updated the figure and also revised the corresponding value in the manuscript.

**Overall, the bias-corrected RF can handle both short and long gaps very well. Specifically, it demonstrates significantly superior performance compared to the MDS method when handling very-long gaps. Additionally, it allows for temporal prolongation, which the MDS method cannot achieve. When compared with original RF, our method can improve the poor performance of RF under very-short and short gap scenario.**

*In revised paper Figure 4, Figure 5:*

[Figure]

***Figure 4*** *The gap-filling performance of three algorithms under different gap-length scenarios. The left panels show the results of the root mean square error (RMSE, W/m²) and the right panels show results of correlation coefficient (CC) between gap-filled values and observations. Different rows of this figure indicate different land cover types. The three horizontal lines of the boxes indicate the first quartile, median, and third quartile, respectively, and the black dots indicate the means. Data labels in this figure are the mean value of RMSE and CC. MDS: marginal distribution sampling. RF: random forest.*

[Figure]

*Figure 5* *The bias between gap-filled values and observations of three methods under different gap-length scenarios. Different rows of this figure indicate different land cover types. The three horizontal lines of the boxes indicate the first quartile, median, and third quartile, respectively, and the black dots indicate the means. Data labels in this figure are the mean value of Bias. MDS: marginal distribution sampling. RF: random forest.*

**4:** Line 185 – Please elaborate more on how you choose the best hyperparameters from 64 models. 64 models with 64 sets of parameters are obtained. For the sites with similar land type, are those models combined into one unified model by taking averages of parameters or still using different sets of parameters? Please explain it in more details.

**Reply:** Thank you for your suggestion.

For each site, the training and test dataset were generated 20 times, so we did the 10-fold cross validation for 20 times and gained 20 hyperparameter combinations. We found that for each site, the 20 hyperparameter combinations are almost the same. Therefore, we choose the hyperparameter combination based on two criteria: (1) achieving optimal model performance, and (2) exhibiting the highest frequency of occurrence across 20 experimental trials. **Consequently, each site has a site-specific and unique hyperparameter combination.**

In general, we trained model and get the best hyperparameter combination **site by site**. We did not unify the models from sites with the same land cover type. We have modified our expression in our revised paper to make it clearer.

*In revised paper Line 185-191:*

*"For each site, the training and test dataset were generated 20 times, so we did the 10-fold cross validation for 20 times and gained 20 hyperparameter combinations. We found that for each site, the 20 hyperparameter combinations are almost the same. Therefore, we choose the hyperparameter combination based on two criteria: (1) achieving optimal model performance, and (2) exhibiting the highest frequency of occurrence across 20 experimental trials. Consequently, each site has a site-specific and unique hyperparameter combination."*

**5:** In discussion section, please add potential limitations from this study in terms of variable importance, sensitivity and stability.

**Reply:** Thank you for your suggestion. We have added a "Advantages and disadvantages" part in discussion section, including the potential limitations from this study in terms of variable importance, sensitivity and stability.

*In revised paper Line 416-427:*

*"5.3 Advantages and disadvantages*

*Our study presents several notable advantages: 1) The bias-corrected RF shows better performance than the official MDS approach, especially for filling very long gaps (up to 30 days). Additionally, it allows for temporal prolongation, which the MDS method cannot achieve. Furthermore, our method enables the incorporation of a broader range of reference variables to establish a more robust non-linear relationship between LE and its drivers; 2) Compared to the FLUXNET2015 dataset, our hourly gap-filled data show improved quality and simpler implementation. The daily prolonged data provide extended temporal coverage (2000-2022) that is particularly valuable for evapotranspiration (ET) modeling and global-scale studies. However, some limitations in terms of variable importance, sensitivity and stability merit further discussion. The variable importance analysis (Section 5.2) indicated our method exhibits strong sensitivity to SW_IN data for gap-filling and to NDVI for prolongation. While we implemented bias correction between ground observations and ERA5-Land data, potential quality issues in SW_IN and NDVI inputs may still affect final results. Future improvements could incorporate higher-quality input data with more stable biases to enhance result reliability."*

**Referee 2:**

**General comment:**

This study presents a 23-year long-term benchmark ET dataset (2000–2022) based on global FLUXNET2015 observations. The dataset effectively addresses critical gaps in existing ET records at both hourly and daily scales, while also extending the time span. This makes it highly valuable for validating ET models and satellite-derived ET products. Therefore, this work has significant importance for the ET research community and is suitable for ESSD. Below are some minor suggestions to further enhance the quality of the manuscript.

**Reply:** Thank you for taking your precious time to review our article and also recognizing our work. We have made a detailed reply and they are as follows. We have also carefully examined the entire manuscript and corrected some ambiguous or incorrect expressions therein. All the changes are marked in red.

**1:** Line 14: "terrestrial"

**Reply:** Thank you for pointing out the mistake. We have modified "the terrestrial" to "terrestrial".

> *In revised paper Line 14:*
> *"Evapotranspiration (ET) is a crucial component of terrestrial hydrological cycle."*

**2:** Line 19: "This hinders their application." the sentence is too short

**Reply:** Thanks for your comment. We have rewritten this sentence to make it more specific.

> *In revised paper Line 17-19:*
> *"However, existing $LE_{EC}$ datasets, like FLUXNET2015, face significant challenges due to limited observation periods and extensive data gaps, which hinders their application in ET modelling and global change analysis"*

**3:** Line 45: "With the abundance of data and the development of models" the sentence is not very clear.

**Reply:** Thanks for your comment. We have rewritten this sentence to make it clarity.

> *In revised paper Line 47-49:*
> *"With the abundance of remotely-sensed and reanalysis data and the development of ET models, more and more ET products based on remote sensing or earth system model simulation are produced and shared."*

**4:** Line 50: "as" is missing in "Since LEEC data are considered…"

**Reply:** Thank you for pointing out the mistake. We have added "as" after "considered".

> *In revised paper Line 53:*
> *"Since $LE_{EC}$ data are considered as the ground truth, researchers are eager to find evidence from ground observations to support their hypotheses."*

**5:** Line 62: "hopes" seems not suitable

**Reply:** Thanks for your suggestion. We have replaced the word to "*aspires*" and rewritten this

sentence.

*In revised paper Line 64-65:*
*"The research community aspires to leverage the most recent, long-term $LE_{EC}$ data; however, there is a lack of up-to-date datasets that are readily accessible for their use."*

**6:** Eq. 2-3: Ta and Td is better than ta and td. And also, ata is easily mistaken for a single symbol.
**Reply:** Thanks for your suggestion. Here, we have changed the symbol of air temperature from 'ta' to '$T_a$' and dewpoint temperature from 'td' to '$T_d$'. The use of 'ata' can indeed lead to confusion. Therefore, we add a multiplication symbol between 'a' and '$T_a$'.

*In revised paper Eq. 2-3:*

$$e_s = 6.1078 \times exp\left(\frac{a \times T_a}{Ta+273.15-b}\right) \begin{cases} a = 17.27, \ b = 35.86, T_a > 0 \\ a = 21.87, \ b = 7.66, T_a \leq 0 \end{cases}, \quad (2)$$

$$e = 6.1078 \times exp\left(\frac{a \times T_d}{T_d+273.15-b}\right) \begin{cases} a = 17.27, \ b = 35.86, T_a > 0 \\ a = 21.87, \ b = 7.66, T_a \leq 0 \end{cases}, \quad (3)$$

*where $e_s$ is the saturated vapour pressure (kPa), $e$ is the actual vapour pressure (kPa), $T_a$ is the air temperature, and $T_d$ is the dewpoint temperature (°C).*

**7:** Line 109: The sentence of "Its spatial resolution…" needs to be refined and polished.
**Reply:** Thanks for your suggestion. We have rewritten this sentence and it is terse now.

*In revised paper Line 111:*
*"Its spatial and temporal resolutions are 250 m and 16 days, respectively."*

**8:** Line 112: replace "contained" by "includes"
**Reply:** Thanks for your suggestion. We have replaced "contained" by "includes".

*In revised paper Line 114:*
*"The gap-filling and prolongation framework for $LE_{EC}$ data mainly includes 3 parts:…"*

**9:** Line 123: "if"-> "when"
**Reply:** Thanks for your suggestion. We have replaced "if" by "when".

*In revised paper Line 124-125:*
*"We calculated the daily energy balance ratio (EBR) when there were ≥36 (18 for hourly data) valid observations in a day."*

**10:** Line 129: What does the sentence of "thus we chose 2 more sites with relatively good data quality" mean?
**Reply:** Thanks for your question. Following our site selection criteria, no African sites initially qualified for inclusion. To enhance regional representation, we incorporated two additional African sites (CG-Tch and SD-Dem) that partially met the requirements. While these sites satisfied Criteria 1 and 3, they exhibited higher data gaps (57% and 64% missing values respectively, see Appendix A: Site information) compared to our 50% threshold. Nevertheless, these sites represent the best available data quality within the region. For clarity, we have reformulated this explanation in the revised manuscript.

*In revised paper Line 130-131:*

*"Notably, no sites in Africa fully met the specified criteria. Consequently, we selected two additional sites that substantially met the essential requirements."*

**11:** Line 212: Why did the author prolong the daily ET? not the hourly ET?

**Reply:** Thanks for your question. The decision to focus exclusively on daily-scale ET prolongation was mainly based on the consideration that **current mainstream ET products are predominantly available at daily and monthly scales** (Zhang et al., 2019; Zheng et al., 2022; Miralles et al., 2025). **Therefore, prolonging daily-scale ET data aligns best with current practical application scenarios. Of course, if required in the future, we can also extend hourly-scale data to meet future research needs**. We have added this explanation in our revised manuscript.

Reference:

Zhang, Y., Kong, D., Gan, R., Chiew, F. H., McVicar, T. R., Zhang, Q., and Yang, Y.: Coupled estimation of 500 m and 8-day resolution global evapotranspiration and gross primary production in 2002–2017, Remote Sens. Environ., 222, 165-182, https://doi.org/10.1016/j.rse.2018.12.031, 2019.

Zheng, C., Jia, L., and Hu, G.: Global land surface evapotranspiration monitoring by etmonitor model driven by multi-source satellite earth observations, J. Hydrol., 613, 128444, https://doi.org/10.1016/j.jhydrol.2022.128444, 2022.

Miralles, D. G., Bonte, O., Koppa, A., Baez-Villanueva, O. M., Tronquo, E., Zhong, F., Beck, H. E., Hulsman, P., Dorigo, W., Verhoest, N. E. C., and Haghdoost, S.: Gleam4: Global land evaporation and soil moisture dataset at 0.1° resolution from 1980 to near present, Sci. Data, 12, 10.1038/s41597-025-04610-y, 2025.

*In revised paper Line 130-131:*

*"Current mainstream ET products are predominantly available at daily and monthly scales (Zhang et al., 2019; Zheng et al., 2022; Miralles et al., 2025). Therefore, prolonging daily-scale ET data aligns best with current practical application scenarios."*

**12:** Line 304: What's the difference between the forward and the backward prolongation?

**Reply:** Thanks for your question.

The difference between the forward and the backward prolongation is two opposite time direction. For example, one site has the time cover from 2007-2014. Therefore, prolongation of 2000-2006 is the backward direction, and prolongation of 2015-2022 is the forward direction. A schematic diagram is shown below. We expect that the prolongation performance will be consistent in both directions. Our analysis confirms that the extension algorithm achieves consistent performance in both directions, meeting our expectations (Figure 7 in the manuscript). We have added the explanation in our revised manuscript.

*In revised paper Line 232-234:*

*"The prolongation at the daily scale was conducted into two time directions: forward and backward. For example, one site has the time cover from 2007-2014. Therefore, prolongation of 2000-2006 is the backward direction, and prolongation of 2015-2022 is the forward direction. We expect that the prolongation performance will be consistent in both directions."*

[Figure]

A schematic diagram for the forward and the backward prolongation.

**13:** For the datasets, what is the difference between the aggregated_daily and the prolonged_daily_200_2022?

**Reply:** Thanks for your question.

**"Aggregated_daily" data** was aggregated from the gap-filled half-hourly data to a daily scale. The start and the end times match the observation period at each site. **"Prolonged_daily_2000-2022" data** provides the prolonged daily data using the novel bias-corrected RF algorithm. The seamless data spans 2000-02-18 to 2022-12-31. For the prolonged part, the quality flag is set to 2. The rest is consistent with the aggregated daily data. In general, **"Aggregated_daily" data** is subset of the **"Prolonged_daily_2000-2022" data**. We have modified the "Data availability" part in our revised paper.

> *In revised paper Line 130:*
> *"6 Data availability*
> *Our released dataset mainly contains four types of data:*
> *(1) Half-hourly or hourly gap-filled data: The data were well gap-filled data using the novel bias-corrected RF algorithm. Filenames include "HH" for half-hourly or "HR" for hourly data. The time format follows FLUXNET2015 standards, with paired timestamps recorded in local time. The start and end times align with the observation period at each site. For QC flags, a value of 0 indicates observed data, while 1 indicates gap-filled data.*
> *(2) Aggregated daily data. This daily dataset was aggregated from the gap-filled half-hourly data to a daily scale. The start and the end times match the observation period at each site. QC flags represent the percentage of valid hourly observations for each day.*
> *(3) Prolonged daily data: This dataset provides the prolonged daily data using the novel bias-corrected RF algorithm. The seamless data spans 2000-02-18 to 2022-12-31. For the prolonged part, the quality flag is set to 2. The rest is consistent with the aggregated daily data.*
> *(4) Aggregated monthly and yearly data: These datasets were aggregated from the prolonged daily data. QC flags indicate the proportion of days with >90% valid hourly observations per month or year. No distinction is made between prolonged data and completely missing daily*

*data. The time span for the monthly data is 2000-03 to 2022-12, and that for the yearly data is 2001-2022.*

*All files are formatted as csv files. NDVI and debiased reference variables from ERA5-Land are also provided in our released data. The product has been deposited at https://doi.org/10.5281/zenodo.13853409 (Li et al., 2024) and can be downloaded publicly."*